# Screening and Isolating Major Aldose Reductase Inhibitors from the Seeds of Evening Primrose (*Oenothera biennis*)

**DOI:** 10.3390/molecules24152709

**Published:** 2019-07-25

**Authors:** Zhiqiang Wang, Shigang Shen, Ze Cui, Hailiang Nie, Dandan Han, Hongyuan Yan

**Affiliations:** 1Key Laboratory of Medicinal Chemistry and Molecular Diagnosis, Ministry of Education & College of Public Health, Hebei University, Baoding 071002, China; 2Key Laboratory of Analytical Science and Technology of Hebei Province, College of Chemistry and Environmental Science, Hebei University, Baoding 071002, China; 3Hebei Provincial Center for Disease Control and Prevention, Shijiazhuang 050021, China

**Keywords:** evening primrose seeds, *Oenothera biennis*, aldose reductase, ultrafiltration, HSCCC

## Abstract

Aldose reductase (AR) is a drug target for therapies to treat complications caused by diabetes mellitus, and the development of effective AR inhibitors (ARIs) of natural origin is considered to be an attractive option for reducing these complications. In this research, the rat lens AR (RLAR) inhibitory activity of evening primrose (*Oenothera biennis*) seeds was investigated for the first time. In our results, the 50% (*v*/*v*) methanol extract of evening primrose seeds exhibits excellent RLAR inhibitory activity (IC_50_ value of 7.53 μg/mL). Moreover, after enrichment of its bioactive components, the ARIs are more likely to be present in the ethyl acetate fraction of 50% (*v/v*) methanol extract (EME) of evening primrose seeds, which exhibits superior RLAR inhibitory activity (IC_50_ value of 3.08 µg/mL). Finally, gallic acid (**1**), procyanidin B3 (**2**), catechin (**3**), and methyl gallate (**4**) were identified as the major ARIs from the EME by affinity-based ultrafiltration-high-performance liquid chromatography and were isolated by high speed countercurrent chromatography, with gallic acid (11.46 µmol/L) and catechin (14.78 µmol/L) being the more potent inhibitors of the four ARIs identified. The results demonstrated that evening primrose seeds may be a potent ingredient of ARIs.

## 1. Introduction

Diabetes mellitus, a disorder characterized by an unnaturally high blood glucose concentration, is a global pandemic that currently affects approximately 415 million adults worldwide [1] and is estimated to affect 552 million adults by 2030 [2]. Diabetic complications (both micro- and macrovascular) induced by sustained long-term hyperglycemia are the primary factors in dysfunction and death of diabetic patients [3,4,5]. Although the development of diabetic complications can be slowed through blood glucose regulation, it cannot be stopped [6,7]. Hence, the occurrence of these complications in the vast majority of diabetic patients is expected [8], and for this reason, exploring novel agents for the prevention and treatment of diabetic complications is desperately needed.

Aldose reductase (AR) is an enzyme that exists in most human cells and is the drug target for therapies to treat complications arising from diabetes. Under conditions of euglycemia, the decomposition of toxic aldehydes generated by lipid peroxidation is the main physiological role of AR [9]. In diabetes mellitus, however, AR becomes the key rate-limiting enzyme of the polyol pathway that catalyzes glucose conversion to sorbitol with the consumption of β-nicotinamide adenine dinucleotide 2′-phosphate reduced tetrasodium salt hydrate (NADPH), resulting in cellular osmolarity and redox imbalances; AR, thus, mediates tissue and vascular damage [10,11]. Plus, several recent studies have demonstrated that AR overexpression can be triggered by hyperglycemia, which can exacerbate the onset of diabetic complications [12]. Therefore, the development of novel and effective AR inhibitors (ARIs) is considered to be an attractive option for alleviating diabetic complications. In recent decades, a vast number of chemical ARIs, such as tolrestat and sorbinil, have been synthesized artificially [13,14,15,16]. However, adverse effects or poor in vivo efficacy have limited their clinical applications [17,18]. Thus, development of ARIs from natural origin with less toxicity is considered to be a more attractive option.

Evening primrose (*Oenothera biennis*) is an herbaceous plant of the family *Onagraceae* that is widely distributed throughout the world’s temperate and subtropical regions. Evening primrose has been used medicinally since ancient times for treatment of various conditions, including eczema, asthma, and rheumarthritis [19,20]. The seed of evening primrose has received widespread attention in food science, medicine, and the cosmetics industry because its oil extracts contains a high content of unsaturated fatty acids (e.g., gamma-linolenic acid, linoleic acid, and oleic acid) and has shown excellent health benefits such as antioxidant, anticancer, anti-inflammation, antiarteriosclerosis, and antiaging properties [21,22]. Moreover, polyphenols extracted from evening primrose seeds have shown remarkable bioactivity [23,24]. However, assessing the AR inhibitory activity of evening primrose seeds has not yet been reported. Thus, in this work, the AR inhibitory effect of evening primrose seeds was investigated for the first time, and its major ARIs were enriched and isolated using high speed countercurrent chromatography (HSCCC) guided by affinity-based ultrafiltration-high performance liquid chromatography (HPLC).

## 2. Results and Discussion

### 2.1. Rat Lens AR (RLAR) Inhibitory Effects of Evening Primrose Seed Extract

Because the activities of the bioactive ingredients in the extracts are influenced by the extraction solvents, *n*-hexane and 50% (*v*/*v*) methanol were evaluated as extraction solvents for their effect on the biological activity of evening primrose seed extracts on the inhibition of RLAR. As shown in Table 1, the 50% (*v*/*v*) methanol extract of the evening primrose seeds exhibited a high inhibitory activity (IC_50_ = 7.53 µg/mL), whereas the *n*-hexane extract showed negligible activity, indicating that the major bioactive ingredients in evening primrose seeds are polar components.

To further enrich these polar bioactive components, the 50% (*v*/*v*) methanol extract of the evening primrose seeds was fractionated by solvent partition with ethyl acetate and water. The inhibitory effects of the ethyl acetate fraction of evening primrose seeds 50% (*v*/*v*) methanol extract (EME) and the water fraction on RLAR were then assessed. As shown in Table 1, the EME exhibited high activity (IC_50_ value of 3.08 µg/mL), indicating that the ARIs are more likely to be present in the EME than in any other fraction. Consequently, the EME was used for further screening.

### 2.2. Screening of Potential ARIs from the EME

In assessing the biological activity of natural products, separation and identification of the active compounds are typically required prior to further evaluation. However, as shown in Figure 1a, the composition of the EME is too complex to isolate and identify the ARIs using conventional methods. Recently, affinity-based ultrafiltration-HPLC has become an alternative popular screening method. In this approach, the complexes of active molecules bound to a drug target can be separated from unbound molecules through ultrafiltration, and the candidate active molecules can then be identified by HPLC analysis; the smaller peaks in the HPLC profile indicate the active molecules bound to the target. In the present study, ultrafiltration-HPLC analysis was performed for rapid screening of ARIs in the EME. However, because the solubility of EME is low in the incubation buffer, the precipitate of the EME buffer solution was removed by centrifugation, and the supernatant of the EME buffer solution (ESME) was collected for further ultrafiltration-HPLC analysis. As shown in Figure 1a,b, although the peak areas of ESME are lower than that of EME, their compositions are almost same. By comparing the chromatograms of Figure 1c,d, the peak areas of compounds **1**, **2**, **3**, and **4** were decreased, indicating that these compounds are potent ARI candidates that should be isolated in subsequent experiments.

### 2.3. HSCCC Separation of Potential ARIs from EME

An appropriate two-phase solvent system that provides a satisfactory partition coefficient (*K*) and separation factor (α) is crucial for HSCCC separation. If the *K* value is less than 0.5 or more than 2, the analytes separated by HSCCC will be eluted quickly close to the solvent front or retained for a long-time elution. Additionally, α values ≥ 1.5 provide good compound resolution [25]. To isolate the candidate ARIs using HSCCC, separation solvent systems comprising *n*-hexane (HE)/ethyl acetate (EA)/methanol (ME)/water (WA) with different volume ratios were examined in this study and their corresponding *K* and α values were evaluated. As shown in Table 2, the HE/EA/ME/WA (1/3/0/4 and 1/5/1/5, *v*/*v*/*v*/*v*) solvent systems did not provide suitable *K* value ranges for the target compounds (**1**, **3**, and **4**) and cannot therefore be applied for HSCCC separation. By comparison, the HE/EA/ME/WA (1.25/5/1.25/5, 1.75/5/1.75/5, and 2/5/2/5, *v*/*v*/*v*/*v*) solvent systems did provide suitable *K* value ranges for the target compounds **2** and **4**, but the α values between some compounds (such as those between **2**, **4**, and **7**, or **3** and **5**) are small. The α values between the targets and other compounds were improved by the HE/EA/ME/WA (1.5/5/1.5/5, *v*/*v*/*v*/*v*) system. Although compound **1** will be retained for a long time in the column during HSCCC separation, as reflected by its high *K* value (3.13), this is considered to be acceptable. Similarly, the relatively low *K* value (0.31) for compound **3** is also considered to be acceptable.

The HE/EA/ME/WA (1.5/5/1.5/5, *v*/*v*/*v*/*v*) solvent system was applied for the HSCCC separation of compounds **1**, **2**, **3**, and **4** from the EME fraction, affording a stationary phase retention ratio of 58%. Figure 2 shows the HSCCC separation chromatogram of the EME; compounds **1** (45.4 mg), **2** (18.3 mg), **3** (8.2 mg), and **4** (58.1 mg) were isolated with purities of 99.1%, 93.3%, 89.7%, and 98.9%, respectively, by HPLC (Figure 3). Compounds **1**, **2**, **3**, and **4** were identified by ^1^H NMR spectroscopy as gallic acid, procyanidin B3, catechin, and methyl gallate, respectively, and their chemical structures are shown in Figure 4. Flavan-3-ol derivatives (catechin and procyanidin B3) and gallic acid derivatives (gallic acid and methyl gallate) are the major bioactive polyphenols in evening primrose seeds. Notably, these compounds have also been isolated or detected in previously reported studies [26].

### 2.4. RLAR Inhibitory Activities of the Isolated Compounds

The AR inhibitory activities of the identified compounds were confirmed by RLAR assays. As shown in Table 3, all four compounds showed inhibitory activities on RLAR at concentrations of 10 µg/mL, which was consistent with our predictions of ultrafiltration-HPLC analysis and previously reported results. The IC_50_ values of gallic acid (**1**) and catechin (**3**) were 11.46 µmol/L and 14.78 µmol/L, respectively, whereas those of procyanidin B3 (**2**) and methyl gallate (**4**) were not obtained. Relative to the positive control (quercetin), the RLAR inhibitory activities of these four compounds are low. Collectively, these results suggest that compounds **1**–**4** are major ARIs of evening primrose seeds and are responsible for its RLAR inhibition; however, the results also suggest that other potent ARIs may be present in evening primrose seeds, and this should be systematically studied in future research.

## 3. Materials and Methods

### 3.1. Reagents and Materials

Human recombinant AR was supplied by Wako Pure Chemical Industries Ltd. (Osaka, Japan). Ammonium sulfate, disodium hydrogen phosphate dodecahydrate, _DL_-glyceraldehyde, NADPH, potassium dihydrogen phosphate, quercetin, sodium dihydrogen phosphate dehydrate, sodium hydroxide, and trifluoroacetic acid were obtained from Sigma-Aldrich (St. Louis, MO, USA). HE, EA, and ME were purchased from J.T. Baker (Phillipsburg, NJ, USA). The deionized WA (resistivity > 18.2 MΩ cm) was produced by a Milli-Q system (Millipore, Bedford, MA, USA) in our laboratory.

### 3.2. Extraction and Fractionation of Evening Primrose Seeds

The dried seeds of evening primrose were supplied by a local market in Anguo, Hebei Province of China. A voucher sample (WLS-EPS-1) has been deposited in the Food Safety Monitoring and Risk Assessment Laboratory, College of Public Health, Hebei University, Baoding, China. The dried seeds (100 g) were first extracted with *n*-hexane (500 mL) for 48 h. They were subsequently filtered using filter paper and dried using a reduced pressure rotary evaporator (RE-2000A, Yarong Co., Ltd., Shanghai, China) at 37 °C to yield the hexane extract (8.37 g). The filter residues were then collected and extracted by 50% (*v*/*v*) methanol (500 mL) for 48 h. The 50% (*v*/*v*) methanolic extract was also filtered and dried using the rotary evaporator (RE-2000A). The concentrated 50% (*v*/*v*) methanolic extract was then freeze-dried to obtain 6.51 g of a 50% (*v*/*v*) methanolic extract solid powder. The 50% (*v*/*v*) methanolic extract solid powder (5 g) was then fractionated through partition with ethyl acetate (200 mL) and water (200 mL) to yield the ethyl acetate and water fractions after the solvents were removed by rotary evaporation (RE-2000A) and freeze drying.

### 3.3. Animals

Sprague–Dawley rats (weight 250–280 g) were supplied by Beijing Vital River Laboratory Animal Technology Co., Ltd. All procedures for the animal tests conformed to the guidelines and approval of the Institutional Animal Care and Use Committee (IACUC-2019004SR) of Hebei University. For acclimatization, the rats were fed with adequate food and water under standardized laboratory conditions (20–26 °C, 40–70% humidity, 12 h light/12 h dark cycle) for one week prior to experimentation.

### 3.4. Determination of RLAR Activity

To extract the RLAR, the eye lens removed from the rats were first homogenized with sodium phosphate buffer (0.1 M, pH 6.2). The resultant homogenate was centrifuged at 10,000× *g* for 30 min at 4 °C. Finally, the supernatant that retained the RLAR was collected for further tests. For assessing the catalytic activities of the RLAR, RLAR (0.65 U/mg), NADPH (0.16 mM), ammonium sulfate (400 mM), _DL_-glyceraldehyde (2.5 mM), and the test samples or quercetin (10–0.1 µg/mL) were mixed in a cuvette, and the NADPH concentrations were monitored by a spectrophotometer (SECOMAM, Ales Cedex, France) at 340 nm for 3 min. The inhibitory activities of the samples can be calculated using Equation (1):(1)$$\mathrm{Inhibition}\;\left(\%\right)=\left(1-\frac{\Delta A_{sample}-\Delta A_{blank}}{\Delta A_{control}-\Delta A_{blank}}\right)\times 100\%$$
where Δ*A_control_* is the absorbance change in 3 min of the tested mixture without the sample; Δ*A_blank_* is the absorbance change in 3 min of the tested mixture without RLAR; and Δ*A_sample_* is the absorbance change in 3 min of the tested mixture. The IC_50_ values of the active samples were also calculated and listed, where IC_50_ indicates the concentration of the tested sample required to consume half of the NADPH.

### 3.5. Ultrafiltration Procedures

The EME was dissolved in the sodium phosphate buffer (0.1 M, pH 6.2). Then, the precipitate was removed by centrifuge, and the ESME was collected for further processes. Briefly, a mixture of ESME, human recombinant AR (6.9 µM), and ammonium sulfate (0.6 M) was first incubated at 37 °C for 30 min and subsequently filtered using the ultrafiltration unit (Microcon YM-10, Millipore, Bedford, MA, USA) and centrifugation at 10,000× *g* for 30 min at 4 °C. The ultrafiltrate was collected for further HPLC analysis. The sample incubated without human recombinant AR was used as the control.

### 3.6. HPLC Conditions

HPLC analysis was performed using an Agilent 1100 HPLC instrument (Agilent, Santa Clara, CA, USA) equipped with an Eclipse Plus C18 column (150 mm length, 4.6 mm i.d., and 5 µm particle size; Agilent, Santa Clara, CA, USA). The temperature of the column oven was set at 30 °C and the injection volumes of the samples were 10 µL. Acidified water (0.1% (*v*/*v*) trifluoroacetic acid, (A) and methanol (B) were used as the mobile phases, and the optimized gradient conditions for the chromatographic separation were 5% (*v*/*v*) eluent B at 0–5 min; 5–100% (*v*/*v*) eluent B at 5–55 min; and 100% (*v*/*v*) eluent B at 55–60 min. The separation was achieved at a flow rate of 0.7 mL/min. The HPLC chromatograms were obtained under 254 nm UV light.

### 3.7. Separation by HSCCC

To evaluate the *K* values for HSCCC separation, the solvent system consisting of HE/EA/ME/WA was selected and used in different ratios (1/3/0/4, 1/5/1/5, 1.25/5/1.25/5, 1.5/5/1.5/5, 1.75/5/1.75/5, and 2/5/2/5, *v*/*v*/*v*/*v*). Each phase of the above solvent system (5 mL) and EME (5 mg) were mixed. After shaking the mixture vigorously for 1 min, two layers were allowed to completely separate. Once two clear layers were formed, 1 mL of each phase was collected and dried; they were then redissolved in methanol (1 mL) for HPLC quantitative analysis. The *K* value of each compound can be calculated through the determined peak area via Equation (2):(2)$$K=\frac{A_{UP}}{A_{LP}}$$
where *A_UP_* is the peak area of a compound detected in the upper phase; *A_LP_* is the peak area of a compound detected in the lower phase. The α value between any two compounds can be calculated through their *K* values via Equation (3):(3)$$\alpha =\frac{K_{1}}{K_{2}}$$
where *K*_1_ and *K*_2_ are the *K* values of compounds 1 and 2, which *K*_1_ should be higher than *K*_2_.

The TBA-1000A HSCCC instrument (Tauto Biotechnique Company, Shanghai, China) was applied for HSCCC separation. The coil column volume of this instrument is 1 L. The maximum rotational speed and injection volume were 1000 rpm and 20 mL, respectively. To isolate the potential ARIs from the EME, the stationary phase (lower phase of HE/EA/ME/WA (1.5:5:1.5:5, *v*/*v*/*v*/*v*)) was first pumped into the coils. After rotating the filled coiled column to 400 rpm, the mobile phase (upper phase of HE/EA/ME/WA (1.5:5:1.5:5, *v*/*v*/*v*/*v*)) was pumped into the coil column in the tail-to-head direction at a flow rate of 4 mL/min. Once the hydrodynamic equilibrium between the two phases was achieved in the column, the EME (500 mg) solution (18 mL, a mixture of HE/EA/ME/WA in 1.5:5:1.5:5, *v*/*v*/*v*/*v*) was injected into the column. The HSCCC chromatogram was recorded by the UV detector at 254 nm. At the end of elution, methanol (1.2 L) was used to wash residuary analytes out of the coil column.

### 3.8. H NMR Spectroscopy of the Isolated Compounds

The structures of the isolated compounds were identified by ^1^H NMR spectroscopy using a Bruker Avance II 600 instrument (Bruker, Billerica, MA, USA). The signals were processed and interpreted using the Bruker DPX 600 MHz (9.4 T) software package.

## 4. Conclusions

In this research, the AR inhibitory effect of evening primrose seeds was investigated for the first time. The EME showed excellent in vitro RLAR inhibitory activity. Moreover, the major ARIs in the EME were screened and isolated by affinity-based ultrafiltration-HPLC and HSCCC and identified as gallic acid (**1**), procyanidin B3 (**2**), catechin (**3**), and methyl gallate (**4**). The results presented herein demonstrate that evening primrose seeds may be a potent ingredient of ARIs, and other more effective ARIs in evening primrose seeds should be systematically studied in future research.

## Figures and Tables

**Figure 1 molecules-24-02709-f001:**
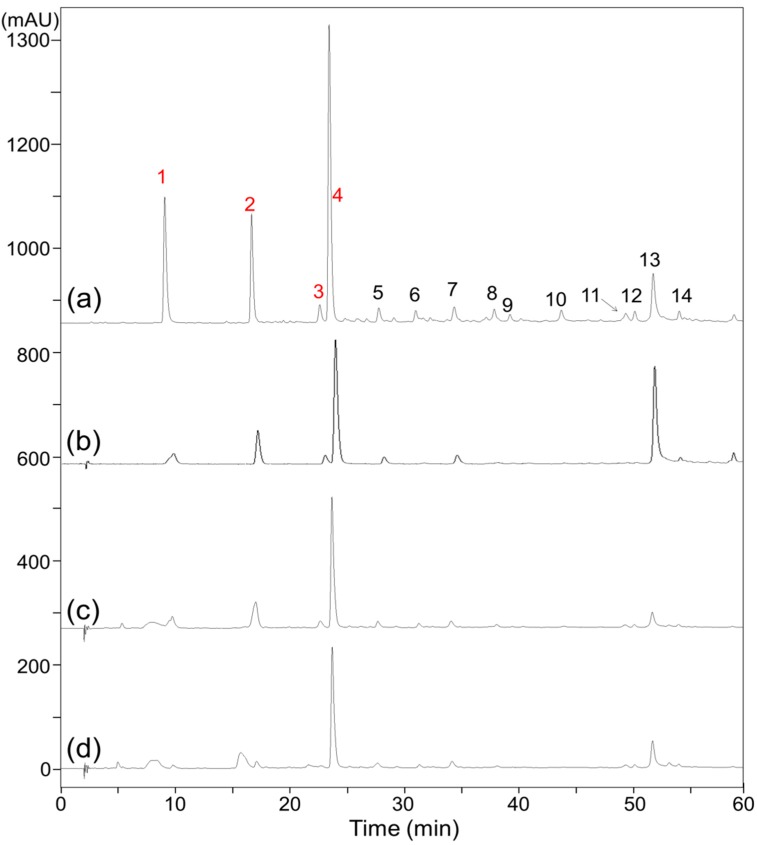
HPLC chromatograms of ethyl acetate fraction of 50% (*v*/*v*) methanol extract of evening primrose seed (EME) (**a**) and supernatant of EME buffer solution (ESME) (**b**), and ultrafiltration-HPLC chromatograms of EME incubated without human recombinant aldose reductase (**c**) and with human recombinant aldose reductase (**d**).

**Figure 2 molecules-24-02709-f002:**
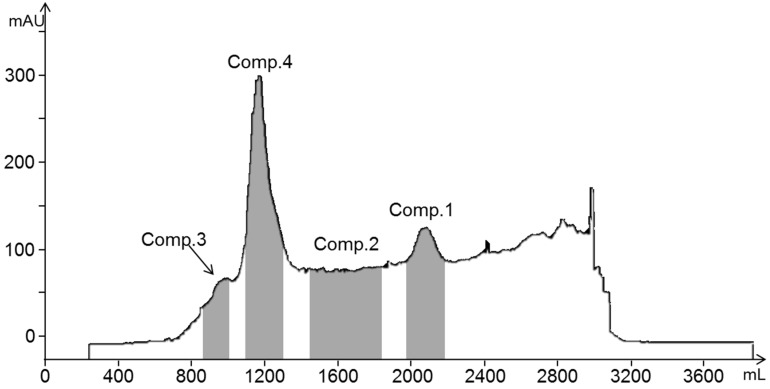
HSCCC chromatogram of the ethyl acetate fraction of the evening primrose seeds 50% (*v*/*v*) methanol extract (EME).

**Figure 3 molecules-24-02709-f003:**
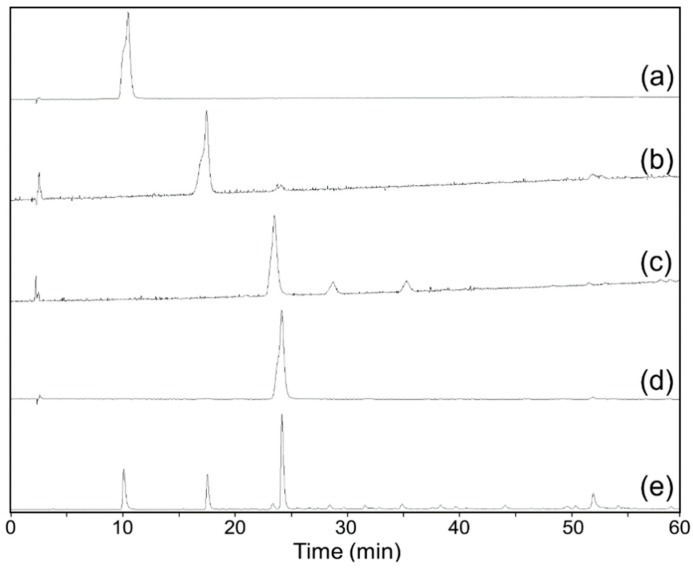
HPLC chromatograms of aldose reductase inhibitors (ARIs) isolated by HSCCC (**a**–**d**) and the ethyl acetate fraction of the evening primrose seeds 50% (*v*/*v*) methanol extract (EME) (**e**). Gallic acid (**a**), procyanidin B3 (**b**), catechin (**c**), and methyl gallate (**d**).

**Figure 4 molecules-24-02709-f004:**
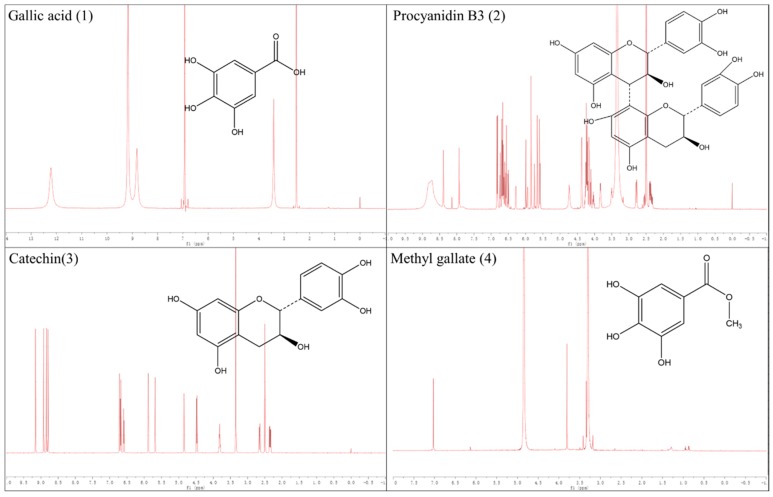
^1^H NMR spectra and chemical structures of target compounds isolated from the ethyl acetate fraction of the evening primrose seeds 50% (*v*/*v*) methanol extract (EME).

**Table 1 molecules-24-02709-t001:** The rat lens aldose reductase (RLAR) inhibitory activities of extracts and fractions from the seeds of *Oenothera biennis.*

Samples	Inhibition (%) ^1^	IC_50_ ^2^
Hexane extract	24.19 ± 8.48 ^4^	- ^5^
50% (*v/v*) Methanol extract	80.88 ± 2.02	7.53 ± 1.67 μg/mL
Ethyl acetate fraction of 50% (*v/v*) methanol extract (EME)	88.58 ± 4.36	3.08 ± 0.84 μg/mL
Water fraction of 50% (*v/v*) methanol extract	46.20 ± 3.24	
Quercetin ^3^	81.57 ± 3.81	0.54 ± 0.13 μg/mL

^1^ The concentration used for RLAR inhibitory assay is 10 μg/mL. ^2^ IC_50_ indicates the concentration of sample necessary to decrease the initial concentration of substrate by 50%. ^3^ Quercetin was used as the positive control. ^4^ Results are presented as mean ± SD (*n* = 3). ^5^ Data was not obtained.

**Table 2 molecules-24-02709-t002:** Partition coefficient values (*K*) of chemical constituents from the ethyl acetate fraction of the 50% (*v*/*v*) methanol extract of evening primrose seeds (EME) in different solvent systems.

Solvent System (*v*/*v*/*v*)	*K* Values of Compounds													
	1 ^1^	2	3	4	5	6	7	8	9	10	11	12	13	14
HE/EA/ME/WA (1/3/0/4)	4.12	0.75	13.26	5.58	0.01	1.41	1.31	6.43	2.84	1.46	4.43	0.13	0.15	0.98
HE/EA/ME/WA (1/5/1/5)	2.23	0.57	0.20	0.44	0.26	3.32	0.30	2.64	2.09	2.92	1.69	1.55	0.39	0.58
HE/EA/ME/WA (1.25/5/1.25/5)	2.41	0.66	0.26	0.57	0.24	4.65	0.51	5.98	7.94	7.89	5.67	4.77	0.35	1.14
HE/EA/ME/WA (1.5/5/1.5/5)	3.13	0.90	0.31	0.66	0.25	8.86	0.22	9.85	12.49	13.85	6.59	4.93	0.44	1.36
HE/EA/ME/WA (1.75/5/1.75/5)	3.59	0.94	0.34	0.73	0.35	8.27	0.22	9.25	12.17	25.61	6.95	5.23	0.48	1.44
HE/EA/ME/WA (2/5/2/5)	4.25	1.11	0.35	0.84	0.37	8.81	0.36	12.06	8.83	12.00	11.09	6.17	0.59	2.31

^1^ Numbering according to Figure 1.

**Table 3 molecules-24-02709-t003:** The rat lens aldose reductase (RLAR) inhibitory activities of isolated compounds from the ethyl acetate fractions of the evening primrose seeds 50% (*v*/*v*) methanol extract (EME).

Samples	Inhibition (%) ^1^	IC_50_ ^2^
Gallic acid (**1**)	80.25 ± 1.56 ^4^	11.46 ± 1.82 μmol/L
Procyanidin B3 (**2**)	44.61 ± 1.99	- ^5^
Catechin (**3**)	69.57 ± 1.99	14.78 ± 3.17 μmol/L
Methyl gallate (**4**)	32.22 ± 0.97	
Quercetin ^3^	81.57 ± 3.81	1.79 ± 0.43 μmol/L

^1^ The concentration used for RLAR inhibitory assay is 10 μg/mL. ^2^ IC_50_ indicates the concentration of sample necessary to decrease the initial concentration of substrate by 50%. ^3^ Quercetin was used as the positive control. ^4^ Results are presented as mean ± SD (*n* = 3). ^5^ Data was not obtained.

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
