# Peer review of "Screening and Isolating Major Aldose Reductase Inhibitors from the Seeds of Evening Primrose (Oenothera biennis)"

_molecules, 2019, doi:10.3390/molecules24152709_

Round 1

Reviewer 1 Report

Dear Sir or Madam,

the manuscript „ Screening and Isolating Major Aldose Reductase Inhibitors from the Seeds Of Evening Primrose (Oenothera biennis)“ describes identification, isolation and characterization of secondary metabolites of evening primrose with the properties of potent aldose reductase inhibitors. The work is done at a good technical level, the manuscript is written in a clear and straightforward way in good English. To my mind, this work is interesting for the community and can be published in Molecules after some revision.

Major remarks

1.      Line 24 and everywhere as applicable: according to SI, the unit of molarity is mol/L, but not M. Thus, µM here needs to be replaced with µmol/L here and analogously everywhere in text.

2.       

Minor remarks

1.      Line 56: “Oil extracts” – I think, here a plural form needs to be used.

2.      Line 67: Why “will be”? I tink, here needs to be “are”

3.      Line 70: What % is meant? I think v/v – am I right?

4.      Line 101: define alpha

5.      Table 2: I think, in the Table legend should be mentioned, that numbering was according Figure 1.

6.      Line 139 and Table 3: write molarities in a correct way please (see major remark 1)

7.      Line 150: legend reference 5 – why the data was not obtained? Could you please give a rationale for this?

8.      Line 164: how to understand – “seeds were condensed”? It is definitely about extract… And why “condensed”? Why not “dried in a rotary evaporator”?

9.      Line 168: “freeze-dried using a freeze-dryer” – what other ways to freeze-dry, besides this?

10.  Line 171: who is the producer of the rotary evaporator?

11.  Line 180: what is “sodium buffer”?

12.  Line 203: what % was used? v/v?

13.  Line 205: …5% ELUENT B

Author Response

Thank you for your kindly comments on our manuscript “Screening and Isolating Major Aldose Reductase Inhibitors from the Seeds of Evening Primrose (Oenothera biennis)”. All the changes in the revision were colored in red and we have made the response for each comment as follows:

Reviewer 1#:

Point 1. Line 24 and everywhere as applicable: according to SI, the unit of molarity is mol/L, but not M. Thus, µM here needs to be replaced with µmol/L here and analogously everywhere in text.

Response 1: Thanks for your comment very much. The “M” was replaced with “µmol/L at lines 24,140, 141, and in the Table 3.

Point 2. Line 56: “Oil extracts” – I think, here a plural form needs to be used.

Response 2: Thanks for your comment very much. The “oil extract” at line 57 was corrected to “oil extracts”.

Point 3. Line 67: Why “will be”? I tink, here needs to be “are”

Response 3: Thanks for your comment very much. The “will be” was changed to “are” at line 68.

Point 4. Line 70: What % is meant? I think v/v – am I right?

Response 4: Thanks for your comment very much. The % at line 71 means v/v, and we comment it at line 71, and all other % in this paper. 

Point 5. Line 101: define alpha

Response 5: Thanks for your comment very much. The α value is the separation factor, and we defined the α value at line 102. Moreover, the calculation method of α value was provided at lines 218-220.

Point 6. Table 2: I think, in the Table legend should be mentioned, that numbering was according Figure 1.

Response 6: Thanks for your comment very much. We indicated “These numbering was according Figure 1” as footnote in Table 2.

Point 7. Line 139 and Table 3: write molarities in a correct way please (see major remark 1)

Response 7: Thanks for your comment very much. As we mentioned in the reply of comment 1, the “M” was replaced with “µmol/L at lines 24, 140, 141, and in the Table 3.

Point 8. Line 150: legend reference 5 – why the data was not obtained? Could you please give a rationale for this?

Response 8: Thanks for your comment very much. As we mentioned in Section 3.4 Determination of RLAR activity, the concentrations of test samples or positive control used for activity test are 10 – 0.1 µg/mL, which 10 µg/mL is a high enough concentration used for activity test in RLAR inhibition assay. Moreover, the IC50 is the concentration of the tested sample that provides 50% inhibition. Thus, if IC50 of one sample want to be obtained, the inhibitions of the sample tested should span 50%. However, in this research, the inhibition of procyanidin B3 (2) and methyl gallate (4) are 44.61% and 32.22% at 10 µg/mL. Thus, the IC50s of procyanidin B3 (2) and methyl gallate (4) cannot be calculated in this case.

Point 9. Line 164: how to understand – “seeds were condensed”? It is definitely about extract… And why “condensed”? Why not “dried in a rotary evaporator”?

Response 9: Thanks for your comment very much. The “condensed” in line 165 and 168 was changed to “dried”.

Point 10. Line 168: “freeze-dried using a freeze-dryer” – what other ways to freeze-dry, besides this?

Response 10: Thanks for your comment very much. The freeze-dryer is the instrument for freeze drying and no other way to do that. We removed “using a freeze-dryer” at line 169.

Point 11. Line 171: who is the producer of the rotary evaporator?

Response 11: Thanks for your comment very much. The producer of the rotary evaporator RE-2000A was Yarong Co., Ltd., Shanghai, China. The producer information was provided when the rotary evaporator was first appeared at line 165. We only provided the producer information one time when the rotary evaporator was first appeared in the paper.

Point 12. Line 180: what is “sodium buffer”?

Response 12: Thanks for your comment very much. The sodium buffer used in RLAR inhibition assay was made by sodium dihydrogen phosphate and disodium hydrogen phosphate. We changed the “sodium buffer” to “sodium phosphate buffer” at line 181.

Point 13. Line 203: what % was used? v/v?

Response 13: Thanks for your comment very much. The % at line 204 means v/v, and we mentioned it at line 204. Furthermore, as we mentioned in the response of comment 4, all other % means v/v were also mentioned in this paper. 

Point 14.  Line 205: …5% ELUENT B

Response 14: Thanks for your comment very much. The proportion of HPLC eluent at line 206-207 was corrected to “5% (v/v) eluent B at 0–5 min; 5–100% (v/v) eluent B at 5–55 min; and 100% (v/v) eluent B at 55–60 min.”

Reviewer 2 Report

The work entitled “Screening and Isolating Major Aldose Reductase 2 Inhibitors from the Seeds of Evening Primrose” describes AR inhibitory effect of EME fraction and isolated compounds derived from seed of evening primrose.

I have major concerns before the manuscript should be considered for submission.

Major concerns

1. In Screening of potential ARIs from the EME section and ultrafiltration procedures, Maybe EME is not soluble in reaction broth with human recombinant AR. So, how to get the UF samples. Isn't this the reason for the decrease in peak area? Because EME is insoluble..

Minor concerns

 In line 93, Figure 2 maybe indicate Figure 1.

Author Response

Thank you for your kindly comments on our manuscript “Screening and Isolating Major Aldose Reductase Inhibitors from the Seeds of Evening Primrose (Oenothera biennis)”. All the changes in the revision were colored in red and we have made the response for each comment as follows:

Reviewer 2#:

Point 1. In Screening of potential ARIs from the EME section and ultrafiltration procedures, Maybe EME is not soluble in reaction broth with human recombinant AR. So, how to get the UF samples. Isn't this the reason for the decrease in peak area? Because EME is insoluble.

Response 1: Thanks for your comment very much. EME is the ethyl acetate fraction of 50% (v/v) methanol extract of evening primrose seeds. Its solubility in reaction buffer is relative low, but not means insoluble. Thus, the ultrafiltration-HPLC experiment could be carried out, and the decreased peak area is the compounds interacted with the human recombinant AR. We will explain these by detailed experimental procedures as follows:

    In order to perform the ultrafiltration assay, a high concentration of EME will be prepared in 50% methanol as a stock solution. Then, dilute EME 50% methanol solution to an appropriate concentration using reaction buffer. During the process of dilution, some components with low solubility of water will be precipitated, but most components will dissolve in the solvent system. The precipitate will be removed by centrifuge, and the supernatant is collected for further incubation (without or with human recombinant AR) and ultrafiltration. As shown in our data of Figure 1 (a), it is the HPLC chromatogram of EME which EME was dissolved in 50% (v/v) methanol, and there are 14 major components can be analyzed by the HPLC chromatogram. As shown in Figure 1 (b), it is the ultrafiltration-HPLC chromatogram of EME incubated without human recombinant AR which the EME used here is the supernatant of reaction buffer solution after centrifugation as we mentioned above. The EME concentration used in in Figure 1 (a) and (b) are same, but the peak areas of Figure 1 (b) are much lower than that of Figure 1 (a). If the EME is insoluble in the reaction broth, there will be no peak in the Figure 1 (b). Furthermore, these peak areas reductions between Figure 1 (a) and (b) are caused by the insoluble components and some components interacted with the membrane of ultrafiltration unit. Thus, the ultrafiltration sample can be obtained.

    Moreover, as we indicated before, as shown in Figure 1 (b), it is the ultrafiltration-HPLC chromatogram of EME incubated without human recombinant AR which the EME used here is the supernatant of reaction buffer solution after centrifugation as we mentioned above. As shown in Figure 1 (c), it is the ultrafiltration-HPLC chromatogram of EME incubated with human recombinant AR which the EME used here is also the supernatant same with in Figure 1 (b). Thus, through comparing the Figure 1 (b) and (c), which the only difference between Figure 1 (b) and (c) is incubation process that with or without human recombinant, and the peak area reductions means the components interacted with human recombinant AR.

    Above all, the ultrafiltration sample can be obtained, and the peak area reduction between Figure 1 (b) and (c) is not because of EME insoluble.

Point 2. In line 93, Figure 2 maybe indicate Figure 1.

Response 2: Thanks for your comment very much. The “Figure 2” was corrected to “Figure 1” at lines 87 and 94.  

Reviewer 3 Report

Congratulations for the work performed. This preliminary study, although submitted as a communication, provides excellent insights for the use of plant-derived bioactives as efficient aldose reductase inhibitors. This is a very interesting work, well written, original and of high scientific interest for readers. It will also be helpful (and I expect exciting to the authors towards to perform more in-depth studies on this field). Only two single aspects should be considered prior acceptance:

- abstract: a conclusion sentence should be included

- methods: why no other extraction solvents were used? and why fractions of the extraction solvents selected were also assessed? what was the fundament for that?

Author Response

Thank you for your kindly comments on our manuscript “Screening and Isolating Major Aldose Reductase Inhibitors from the Seeds of Evening Primrose (Oenothera biennis)”. All the changes in the revision were colored in red and we have made the response for each comment as follows:

Reviewer 3#:

Point 1. abstract: a conclusion sentence should be included.

Response 1: Thanks for your comment very much. The conclusion sentence was added in the abstract as follows: The results demonstrated that evening primrose seeds may be a potent ingredient of ARIs.

Point 2. methods: why no other extraction solvents were used? and why fractions of the extraction solvents selected were also assessed? what was the fundament for that?

Response 2: Thanks for your comment very much. In this work, the evening primrose seeds was extract by n-hexane and 50% methanol, respectively. As we mentioned in the introduction section, the seed of evening primrose has received widespread attention in food science, medicine, and the cosmetics industry because its oil extracts contain a high content of unsaturated fatty acids (e.g., gamma-linolenic acid, linoleic acid, and oleic acid) and has shown excellent health benefits such as antioxidant, anti-cancer, anti-inflammation, anti-arteriosclerosis, and anti-aging properties. Moreover, some studies also reported that polyphenols extracted from evening primrose seeds have also shown remarkable bioactivity. Thus, we used n-hexane as extraction solvent to extract the oil components, and used 50% methanol to extract the polyphenol components. The AR inhibitory activities of these two extract were compared, and the results showed that 50% methanol extract showed better activity, indicating that the AR inhibitory activity of evening primrose seeds is mainly contributed by polyphenols but not oil components.

    The 50% methanol extract of evening primrose seeds was further fractionated by ethyl acetate and water in this work. The meaning of fractionation here is just separation that make the extract simple. As we known, the composition of crude extract is generally too complex to analysis, thus we want to enrich the ARIs by a simple fractionation method for further ultrafiltfation-HPLC analysis and HSCCC separation.

    About the AR inhibitory activity test of evening primrose seeds and ARI isolation, this research is very preliminary, as we mentioned in the paper that compounds 1–4 (gallic acid, procyanidin B3, catechin, and methyl gallate) are major ARIs of evening primrose seeds and are responsible for its RLAR inhibition; however, the results also suggest that other potent ARIs may be present in evening primrose seeds, and this should be systematically studied in future research (actually we are doing now.)

Round 2

Reviewer 2 Report

Thanks for your response about my query. However, I still have a few questions.

 1. You explained detail about your procedures. If so, EME(EA faction of 50% methanol extract) have to change to ESME(EA fraction supernatant of 50% methanol extract). Two sample is not same. The ARI data got from ESME, not from EME.

 2. In figure 1, you have to show a ESME chromatogram definitely and compare another data with it. in these case, figure 1-a is unnecessary

3. Please show the decrease content of peak 1~4, Does it shows statistical significance?

Author Response

Thank you for your kindly comments on our manuscript “Screening and Isolating Major Aldose Reductase Inhibitors from the Seeds of Evening Primrose (Oenothera biennis)”. All the changes in the revision were colored in red and we have made the response for each comment as follows:

Reviewer 2#:

Point 1. You explained detail about your procedures. If so, EME(EA faction of 50% methanol extract) have to change to ESME(EA fraction supernatant of 50% methanol extract). Two sample is not same. The ARI data got from ESME, not from EME.

Response 1: Thanks for your comment very much. The ARI data was got from ESME, and we agree with you that EME and ESME are not same sample. Thus, we introduce the concept of ESME into our manuscript, and the procedures and discussion about ultrafiltration were revised at lines 94-98, 199-201.

Point 2. In figure 1, you have to show a ESME chromatogram definitely and compare another data with it. in these case, figure 1-a is unnecessary

Response 2: Thanks for your comment very much. The chromatogram of ESME was provided in Figure 1 as Figure 1 (b). As we mentioned in the response 1, we agree with you that EME and ESME are not same sample, and so we introduce the concept of ESME into our manuscript. However, we did not remove Figure 1 (a) from our manuscript. Although Figure 1 (a) is less meaning to ultrafiltration-HPLC analysis, but we still need it to compare with ESME to confirm the compositions of EME and ESME are almost same, which is meaning for guiding further HSCCC isolation.

Point 3. Please show the decrease content of peak 1~4, Does it shows statistical significance?

Response 3: Thanks for your comment very much. The peak area reductions of peak 1-4 were 76.67%, 59.83%, 52.31% and 3.21%, respectively. From these data, it looks like that peaks 4 does not showed the statistical significant. However, there are many factors can influence the peak area reductions, such as the concentration ratio between enzyme and extract, as discussed in our previous publications (Journal of Chromatography B, 2015, 1002: 319-328; Journal of Chromatography B, 2017, 1048: 30-37.). Sometimes, the peak area reductions were used for ranking the binding affinities (which can represent the activities), but the conditions of ultrafiltration-HPLC process should be optimized. In this preliminary research, we just screened the ARIs by a simple ultrafiltration-HPLC assay for guiding HSCCC isolation, without optimizing the experiment conditions, and so just identify the hits by simply comparison. Moreover, due to the ultrafiltration-HPLC conditions were not optimized in this preliminary work, the peak area reductions (76.67%, 59.83%, 52.31% and 3.21%) were not exactly matched with activity data (in Table 3).  Therefore, we think the decrease content is not necessary for this preliminary research.
